# Nowcasting thunderstorm hazards using machine learning: the impact of data sources on performance

Jussi Leinonen[1], Ulrich Hamann[1], Urs Germann[1], and John R. Mecikalski[2]

[1]Federal Office of Meteorology and Climatology MeteoSwiss, Locarno-Monti, Switzerland
[2]Atmospheric Science Department, University of Alabama in Huntsville, Huntsville, Alabama, USA

**Correspondence:** Jussi Leinonen (jussi.leinonen@meteoswiss.ch)

**Abstract.** In order to aid feature selection in thunderstorm nowcasting, we present an analysis of the utility of various sources of data for machine-learning-based nowcasting of hazards related to thunderstorms. We considered ground-based radar data, satellite-based imagery and lightning observations, forecast data from numerical weather prediction (NWP) and the topography from a digital elevation model (DEM), ending up with 106 different predictive variables. We evaluated machine-learning models to nowcast storm track radar reflectivity (representing precipitation), lightning occurrence, and the $45\,\mathrm{dBZ}$ radar echo top height that can be used as an indicator of hail, producing predictions for lead times up to $60\,\mathrm{min}$. The study was carried out in an area in the northeast United States, where observations from the Geostationary Operational Environmental Satellite-16 are available and can be used as a proxy for the upcoming Meteosat Third Generation capabilities in Europe. The benefits of the data sources were evaluated using two complementary approaches: using feature importance reported by the machine learning model based on gradient boosted trees, and by repeating the analysis using all possible combinations of the data sources. The two approaches sometimes yielded seemingly contradictory results, as the feature importance reported by the gradient boosting algorithm sometimes disregards certain features that are still useful in the absence of more powerful predictors, while at times it overstates the importance of other features. We found that the radar data is overall the most important predictor. The satellite imagery is beneficial for all of the studied predictands, and therefore offers a viable alternative in regions where radar data are unavailable, such as over the oceans and in less-developed ares. The lightning data are very useful for nowcasting lightning but are of limited use for the other hazards. While the feature importance ranks NWP data as important input, an omission of NWP data can be well compensated by information in the observational data over the nowcast period. Finally, we did not find evidence that the nowcast benefits from the DEM data.

## 1 Introduction

Thunderstorms regularly cause significant risk to human life and damage to property through lightning, heavy precipitation, hail and strong winds. These hazards are highly localized and develop within time scales ranging from tens of minutes to a few hours, which makes them difficult to forecast precisely using numerical weather prediction (NWP) models. NWP models can typically forecast a general tendency for thunderstorms in a given region, but not exactly where and when the most severe

impacts will occur. Thus, issuing localized short-term warnings of impacts is better achieved with *nowcasting*, statistical prediction of near-term (0–1 hour) developments based on the latest available data, in particular observations.

Various tracking and nowcasting systems for thunderstorms have been developed over the years since the 1960s, usually primarily using radar but sometimes also combining other information such as lightning detection and location data. One particularly widely used radar-based system is Thunderstorm Identification, Tracking and Nowcasting (TITAN; Dixon and Wiener, 1993), which tracks thunderstorms as objects defined as continuous regions of high radar reflectivity. A review of other methods developed before 1998 was given by Wilson et al. (1998). More recent radar-based approaches include the combined radar and lightning tracker of and the radar-based algorithms Cell Model Output Statistics (CellMOS; Hoffmann, 2008), TRACE3D (Handwerker, 2002), Thunderstorm Radar Tracking (TRT; Hering et al., 2004, 2005, 2006) and NowCast-MIX (James et al., 2018), while Steinacker et al. (2000) used radar and lightning data in combination. Other algorithms are designed to utilize satellite data instead; prominent examples of these include GOES-R Convective Initiation (Mecikalski and Bedka, 2006; Mecikalski et al., 2015), the Rapid Developing Thunderstorm (RDT; Autonès and Claudon, 2012) algorithm of the Nowcasting Satellite Application Facility (NWCSAF), Cb-TRAM (Zinner et al., 2008; Kober and Tafferner, 2009) and the work of Bedka and Khlopenkov (2016) and Bedka et al. (2018).

Like many other statistical data analysis and prediction tasks, nowcasting of thunderstorms and related hazards has benefited from the rapid advances in machine learning (ML) techniques in the last decade. ML has been popular for nowcasting precipitation (e.g. Shi et al., 2015, 2017; Foresti et al., 2019; Ayzel et al., 2020; Kumar et al., 2020; Franch et al., 2020), and has also been used to develop nowcasting methods for lightning (Mostajabi et al., 2019; Zhou et al., 2020), hail (Czernecki et al., 2019; Huang et al., 2019) and windstorms (Sprenger et al., 2017; Lagerquist et al., 2017, 2020). However, studies so far have typically used only one data source, though in some cases several are utilized. Furthermore, most studies concentrate on predicting only one variable. The variety of adopted methodologies complicates comparisons between the results from different studies.

In this study, our objective is to provide a systematic assessment of the value of various data sources for nowcasting hazards caused by thunderstorms using a ML approach. As a particular goal, we seek to understand the impact on thunderstorm nowcasting from the new generation of geostationary satellites, which, compared to the previous generation, provide higher-resolution imagery, additional image channels and lightning data. Of these satellites, Geostationary Operational Environmental Satellite (GOES) -16 and -17 are currently operational, while the first of the Meteosat Third Generation (MTG) satellites is expected to launch in 2022. Therefore, we conduct our study in the Northeastern US, where the climate is similar to Central Europe (the primary focus of research at MeteoSwiss), and where GOES-16 has a clear field of view. We include a variety of ground-based, satellite-based and model-derived data sources that are available for that region, and examine their value for nowcasting thunderstorms. We base our study on interpretable ML using gradient boosting methods. With our results, we aim to provide guidelines for further research and development such that the investigators can acquire and process the most relevant data sources and variables for their particular applications. Our approach is similar to that of Mecikalski et al. (2021), but we complement that study with a larger number of samples (approximately 88000 vs. 2000), the use of gradient boosting rather than random forests, the inclusion of NWP and digital elevation model (DEM) data, and a more detailed analysis by excluding combinations of different data sources.

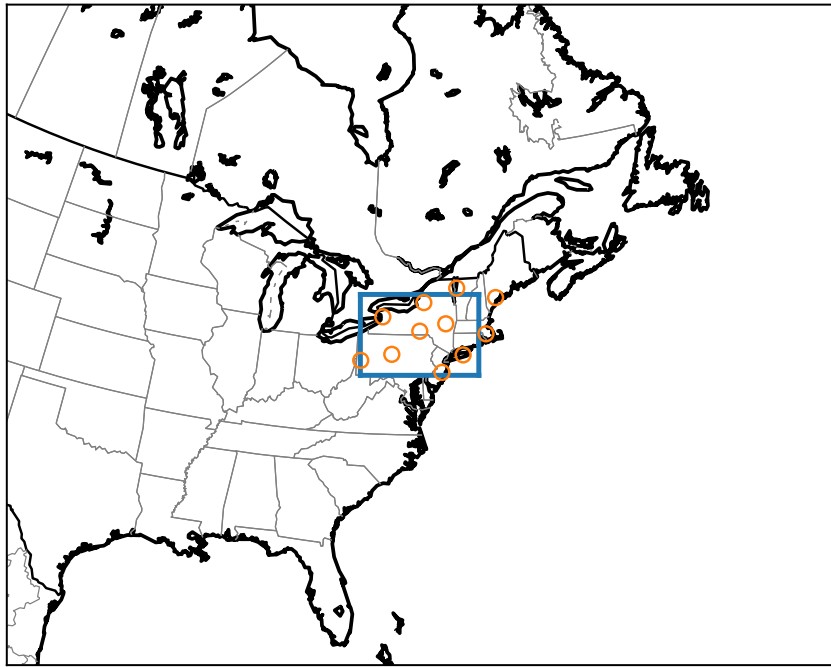

**Figure 1.** The study area in eastern North America. The blue rectangle indicates the area (720 km × 490 km) while the orange circles mark the locations of the NEXRAD radars.

This article is organized as follows: Sect. 2 describes the study region and the data sources, Sect. 3 explains the data processing and ML methods used, Sect. 4 presents the results with discussion of their meaning. Finally, Sect. 5 summarizes and synthesizes the results and their implications for future studies, concluding the article.

## 2 Data

### 2.1 Study area and period

Considering the objectives of the research, we chose to focus on a study area in the northeast of the US, shown in Fig. 1. The study region is a rectangle in azimuthal equidistant projection (Snyder, 1987), centered at 76° W, 42° N with an extent of 720 km in the west–east direction and 490 km in the north–south direction. The resolution of the grid is 1 km per pixel.

The area is centered on the states of New York and Pennsylvania and also covers parts of the states of Connecticut, Massachusetts, New Hampshire, New Jersey, Rhode Island and Vermont, as well as a region of the Atlantic Ocean and a part of the Canadian province of Ontario. Although this region is not as convectively active as, for example, the US Great Plains or the Southeastern US, we chose it because it still experiences considerable thunderstorm activity and the hazard profile of these storms is similar to Central Europe: tornadoes are relatively uncommon, and the hazards consist mostly of hail, lightning, wind

gusts and heavy precipitation (Kelly et al., 1985; Changnon, 1993; Yeung et al., 2015). The latitude of the region is also similar to Central and Southern Europe, and consequently the profiles of solar radiation and the view angles of satellite instruments on geostationary orbit are similar. The main difference between this region and Central Europe is the topography: much of Central Europe is characterized by the Alps, while our study area is generally smoother and most of the variation in elevation is due to the less-prominent Appalachian mountain range.

We collected data from data archives for the period ranging from April to September 2020, with a time resolution of up to 5 minutes depending on the source. The length of the study period and the size of the area were determined as a compromise between gathering an extensive dataset with a large number of samples, while keeping manageable the amount of data (already around 7 terabytes of raw data) that needed to be downloaded and processed.

## 2.2 Data sources

Since the objective of the study was to investigate the utility of different types of data for nowcasting severe thunderstorms, we selected multiple qualitatively different data sources for analysis. In order to constrain the complexity of the study, we tried to avoid unnecessary overlap between the sources; thus, for example, we did not attempt to use similar data from multiple satellites, nor did we obtain ground-based lightning data as that was already available from a satellite source. Moreover, in order to avoid the complications of data intermittency, we preferred to focus on data sources that are regularly available, and avoid sources such as low-Earth orbiting satellites that typically pass over a given area only 1-2 times per day. The final dataset includes data from a ground-based operational radar network, multi-spectral imagery and lightning data from a geostationary satellite, NWP, and DEM data. The details for obtaining the data can be found under "Code and data availability" at the end of the article. The data sources are described in more detail in the following sections.

### 2.2.1 Radar data: NEXRAD

The Next-Generation Radar (NEXRAD; Heiss et al., 1990) network is the US operational radar network operated by the National Weather Service (NWS). It consists of S-band Doppler weather radars that cover most of the continental US as well as many other regions of the country. NEXRAD observations from multiple radars are processed by the National Severe Storms Laboratory (NSSL) into composite products using the Multi-Radar/Multi-Sensor System (MRMS; Zhang et al., 2016; Smith et al., 2016). Unfortunately, the MRMS data are currently only available in near-real time and not publicly archived for more than 24 hours. Therefore, we needed to process the data from individual radars, whose data are publicly archived on the long term, into a composite ourselves; the PyART library (Helmus and Collis, 2016) was used for this purpose. Although this solution has the drawback that we cannot expect to match the quality of a well-developed composite product within this study, an advantage is that using the full three-dimensional measured radar observations allows to calculate any radar variable rather than just those that are available from MRMS. In this work, we derived the column maximum reflectivity (MAXZ), echo top heights at threshold reflectivities of $25\,\mathrm{dBZ}$, $35\,\mathrm{dBZ}$ and $45\,\mathrm{dBZ}$, as well as the vertically integrated liquid (VIL), calculated

as

$$\text{VIL} = 3.44 \times 10^{-6}\, Z^{4/7} \tag{1}$$

where $Z$ is the radar reflectivity given in $\text{mm}^{-6}\,\text{m}^{-3}$ (i.e., $Z = 10^{Z_{\text{dB}}/10}$ for $Z_{\text{dB}}$ in dBZ units) and VIL is in units $\text{kg}\,\text{m}^{-2}$ (Greene and Clark, 1972). The radar data have a time resolution of $5\,\text{min}$.

We selected radars in order to provide good data coverage throughout the study area. The parts of the area that are over ocean and in Canada are within the range of the selected radars, and the entire region is covered with a minimum beam altitude of at most 6000 ft (1800 m), less than 3000 ft (900 m) in most of the region. NEXRAD radars operate using rather shallow scan elevation angles of $0.5°$–$19.5°$ and consequently each individual radar is blind to the region of the atmosphere directly above it. This gap must be filled with nearby radars and therefore we also selected some radars outside the study area in order to ensure adequate 3D data availability within the area. The radars used for the study are listed in Table A1, and their locations are also shown as orange circles in Fig. 1.

### 2.2.2 Satellite imagery: GOES ABI

GOES-16 is a new-generation geostationary satellite with advanced instruments for weather observations (Sullivan, 2020). The primary GOES-16 instrument used in this study is the Advanced Baseline Imager (ABI), which includes 16 bands with wavelengths ranging from $470\,\text{nm}$ (visible) to $13.3\,\mu\text{m}$ (thermal infrared), with resolutions from $0.5$ to $2\,\text{km}$ per pixel in optimal viewing conditions. GOES-16 is located over the equator at $75.2°\text{W}$, a longitude near the middle of our study area. The ABI provides a full-disk scan, a variable region of interest (used for hurricanes, for example), and a scan covering only the contiguous US (CONUS) region. For this study, we use the CONUS scan, which is available at a five-minute time resolution.

We downloaded the level 1 (L1) data (given as reflectance or brightness temperature) for the GOES-16 ABI channels (Schmit and Gunshor, 2020) as well as the level 2 (L2) cloud products cloud top height, cloud top pressure and cloud optical depth (Heidinger et al., 2020) and the derived stability indices (DSI) product (Li et al., 2020), which includes retrievals of variables such as the convective available potential energy (CAPE). We would have preferred to use the cloud top temperature product as well, but it is available only as a full disk product and not separately for the CONUS region. Consequently, we omitted the cloud top temperature because the inferior time resolution (10 min) of the full-disk product would have caused compatibility problems. We also computed the differences of various L1 channels (listed in Table A3) in order to provide better features; see, for example, Mecikalski et al. (2010) for interpretations of the channel differences from geostationary visible/infrared imagers. The data was projected to our study grid with the PyTroll libraries (Raspaud et al., 2018), and corrected for parallax shift using the L2 cloud top height product to determine the appropriate correction.

### 2.2.3 Lightning data: GOES GLM

The GOES-16 satellite is also equipped with the Geostationary Lightning Mapper (GLM), which detects lightning strikes (Rudlosky et al., 2020). The GLM L2 data consists of the coordinates and properties, such as energy, of individual strikes. Each strike consists of multiple lightning "events", which are pixel-level detections of lightning; a set of adjacent and simultaneous

events is interpreted as a strike. The coordinates and properties of the events are also provided, thus providing information about the spatial extent of each lightning strike.

We projected the data of the lightning strikes and events, as well as their energies, to the common grid. The original GLM files contain 20 s of data each, but the files were aggregated such that we create the derived products at 5-min time resolution. The GLM L2 data are provided with parallax correction already performed, obviating the need for this step.

### 2.2.4 Numerical weather prediction: ECMWF

We provide the nowcasting system information about the state of atmosphere in the study area using the NWP products from the integrated forecast system (IFS) of the European Center for Medium-Range Weather Forecasting (ECMWF). We chose to use the global IFS rather than a local-area modeling system as our NWP data source because, unlike the satellite and radar data, it is not limited to a particular region, and we expected this to facilitate the later adaptation of our methodology and results to Europe and other regions beyond the current study area. We obtained a collection of 59 different variables provided by ECMWF; the variables are listed in Table A5.

We use the ECMWF archived forecast product rather than the analysis product in order to only use data that would be available to an operational nowcasting system. We downloaded the ECMWF forecasts at intervals of 12-hours in forecast time, and the data in the forecasts have a 1-hour resolution. To each 5-min time step in our common spatiotemporal framework, we assigned the closest 1-hour time step from the most recently issued forecast. ECMWF provides the data on a latitude–longitude grid; we used the PyTroll tools to project them to our study grid.

### 2.2.5 Digital elevation model: ASTER

Orography can affect the development of convective storms. In order to enable the nowcasting system to exploit information about the elevation and morphology of the terrain, we obtained the Advanced Spaceborne Thermal Emission and Reflection Radiometer (ASTER) global DEM version 3 (Abrams et al., 2020). The resolution of the ASTER DEM is 30 m (the data are provided at a resolution of 1 arcsecond), much finer than our grid pixel size of 1 km. This allows the computation of subpixel properties of the elevation for each grid point. We computed the mean elevation, the elevation gradients and the surface roughness, defined as the root-mean-square (RMS) deviation from the mean, for each pixel in our grid. As a combined variable, we also compute the upslope flow $s$, defined as a dot product of the elevation gradient and the flow velocity:

$$s = \nabla h \cdot \boldsymbol{v} \tag{2}$$

where $h$ is the elevation and $\boldsymbol{v}$ is the flow velocity, which in this case is derived from the radar motion vectors. A positive $s$ indicates that the air flows predominantly uphill, while a negative $s$ corresponds to downhill flow. When we later discuss the importance of data sources in Sects. 4.2 and 4.3, we consider the information content of the upslope flow as one of the DEM variables.

# 3 Methods

## 3.1 Data processing

In order to keep the conclusions of the study general and applicable to different operational environments, we have used general and widely available methods, rather than a particular operational nowcasting system, in the data processing workflow used in this study. It starts with identifying thunderstorm centers in the MAXZ field. The motion of these centers is then tracked
backward and forward in time in a Lagrangian framework by integrating the velocity field obtained with the optical flow method. Once the motion of the center has been estimated, features from different data sources and variables are extracted from the neighborhood of the center at each time step. These features are collected in the ML dataset that is used to train a gradient boosting model. Below, we describe each step of the workflow in more detail.

### 3.1.1 Extraction of storm centers and tracks

After processing the data into a single grid as described in Section 2.2, we identified regions of active thunderstorms in the data based on the observed radar reflectivity. For each 5-minute time step in the data, we located centers of convective activity using the following procedure.

1. Start with an empty list of storm centers

2. Find the pixel with the highest MAXZ, denoted as $p_{\mathrm{maxZ}}$

3. If the MAXZ at $p_{\mathrm{maxZ}}$ is at least 37 dBZ:

   – Add $p_{\mathrm{maxZ}}$ to the list of storm centers.

   – Identify the 25-km diameter circular area surrounding $p_{\mathrm{maxZ}}$ and exclude the pixels in it from further search.

   – Restart the search from step 2.

   Otherwise, end the search.

Thus, storms were identified as regions of high radar reflectivity. We chose the 37 dBZ threshold, which corresponds to a convective precipitation rate of approximately $8 \mathrm{~mm\,h^{-1}}$, following several previous studies in which thunderstorms had been identified by radar reflectivity thresholds of 30–40 dBZ (Marshall and Radhakant, 1978; Wilson and Mueller, 1993; Roberts and Rutledge, 2003; Mueller et al., 2003; Hering et al., 2004; Kober and Tafferner, 2009). To prevent radar artifacts from being identified as storms, we discarded centers that had a valid MAXZ in fewer than $1/3$ of the pixels in the surrounding 25-km
circle.

Once the centers had been identified, we tracked their movement in the domain so that the temporal evolution of the storm is separated from its movement. To estimate the motion, we computed the motion vectors of the reflectivity using the autocorrelation-based optical flow method implemented in the PySteps package (Pulkkinen et al., 2019). This method yields a single motion vector; to allow the motion vector field to vary spatially, we computed a motion vector in this manner for each

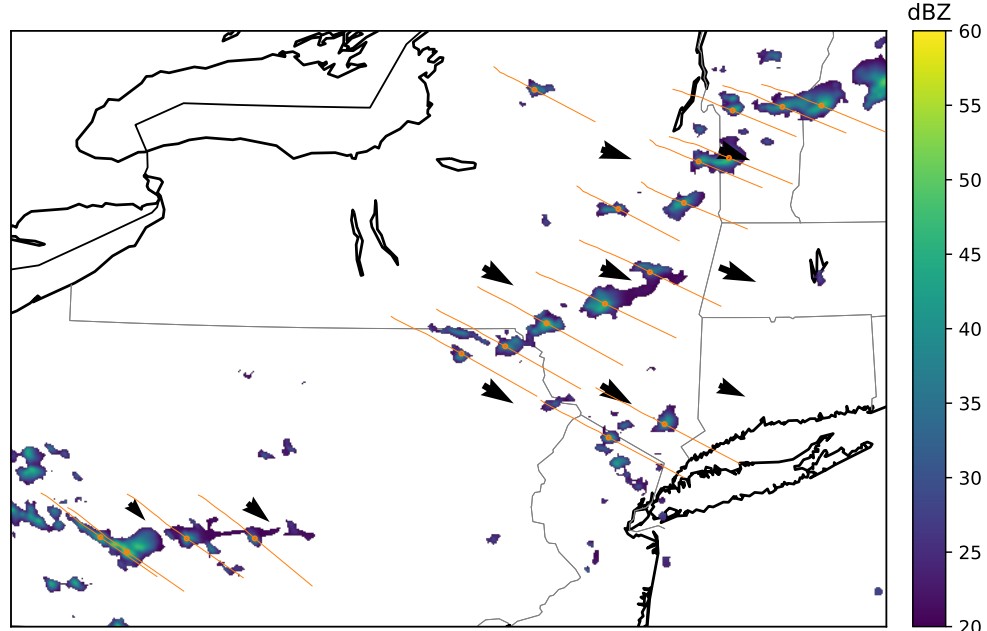

**Figure 2.** An example of the extracted centers at time $t = 0$ (orange circles) and tracks from $t-60$ min to $t+60$ min (orange lines). MAXZ in dBZ is shown in the colored map, with coastlines and state borders in the background.

point in a square grid with a spacing of 97 pixels, using the $200 \times 200$ pixel MAXZ neighborhood of each grid point to compute the vector using the autocorrelation-based method. Once computed in this fashion, the motion vectors were then interpolated to the storm centers. This method produces motion fields with very smooth gradients and is likely to fail to produce the correct motion for regions with high wind shear. Although more advanced methods are available in PySteps, we found these to be more prone to producing artifacts. The procedure described above is more robust and therefore we found it more suitable for the task, required in this study, of automated analysis of tens of thousands of samples.

For each center, we estimated the past location of the corresponding air parcel by backward integrating the motion vectors using Heun's method (also known as improved Euler's method; Süli and Mayers, 2003). On each time step, the advected center may be adjusted by up to 2 pixels to align it at the maximum MAXZ in the neighborhood (we found that 2 pixels was sufficient, and that larger adjustments sometimes caused the tracking to drift to the wrong storm center). For the future motion, only the data that would be available in a real-time nowcasting scenario was used, and therefore we computed the future tracks using the last available motion vectors at the reference time. Both the past and the future tracks are computed for 60 minutes from the reference time. Any tracks that extended out of the study area were discarded. An example of centers and tracks extracted in this manner is shown in Fig. 2.

The storm identification and tracking scheme was implemented with the objectives of robustness and suitability for ML. Therefore, we opted not to use, for instance, the thunderstorm radar tracking (TRT) cell identification method (Hering et al.,

2004, 2005, 2006) which produces variable-sized storm cells, which complicates analysis. Our scheme approximates the tracking of storm centers but may not always perfectly correspond to it. Therefore, one can state the objective of the ML prediction task more precisely as follows: *given the Lagrangian history of a storm centerpoint, selected based on a $37$ dBZ reflectivity threshold, predict its future Lagrangian evolution.*

### 3.1.2 Feature extraction

The evolution of a storm in time is described by the change in the variables in the circular neighborhood of the center. For each variable derived from the data sources described in Section 2.2, we extracted the neighborhood mean, standard deviation and the 10th and 90th percentiles. The percentiles are intended as a soft minimum and a soft maximum, less sensitive to outliers compared to taking the exact minimum and maximum. For Boolean variables such as lightning event occurrence, we also computed binary features that are $1$ if the variable was true at any pixel in the neighborhood and $0$ otherwise.

### 3.1.3 Datasets

The final dataset, collected from the entire study period and study area, comprises $87626$ samples that describe the history and future of the detected storm centers. We divided the samples into a training set that is used to train the ML algorithm, a validation set that is used to evaluate the generalization ability during training, and a test set that is used for final evaluation. We found that simply sampling these sets randomly from the data made the training prone to overfitting because storm tracks found at a similar time and location have similar evolution and thus are not independent samples. In order to improve the independence of the training, validation and testing sets, we determined these sets such that the data from each day (00–24 UTC) were assigned entirely to only one of these sets, mostly eliminating the overlap between them. We sampled the days randomly until at least $10\%$ of the data were in the validation set and at least another $10\%$ in the test set, and assigned the remaining data to the training set. The final datasets are made up of $69594$ training samples, $9160$ validation samples and $8872$ testing samples.

## 3.2 Prediction tasks

The predictands (i.e. the targets of the ML prediction) evaluated in this study were selected based on their relevance for thunderstorm hazard prediction. We examined qualitatively different predictands in order to assess the differences in the contributions of various data sources to the prediction performance.

The first prediction task we define is the nowcasting of the evolution of the column maximum reflectivity on the storm track. This variable is highly indicative of thunderstorm development and can function as an indicator of heavy precipitation and hail. In particular, the radar reflectivity $Z$ can be approximately related to the rain rate $R$ by a relation of the form $Z = aR^b$ where $a$ and $b$ are empirically determined constants. Hereafter, we refer to the task of predicting the evolution of the column maximum reflectivity as MAXZ. We examine the prediction performance for lead times between $5$ and $60$ min.

Another important thunderstorm hazard that we can quantify using the available dataset is the occurrence of lightning. We use the GLM measurements to identify lightning, and approach lightning prediction as a binary task of predicting whether or not lightning will be present in the 25-km diameter neighborhood of the storm center during a given time period. We refer to this task in short as LIGHTNING-OCC.

For hail, we do not have direct observations of its occurrence. However, the presence of hail has been found to be well indicated by the height difference of the radar $45\,\mathrm{dBZ}$ echo top and the freezing height (Waldvogel et al., 1979; Foote et al., 2005; Barras et al., 2019). Since the freezing level is obtained from NWP data, the principal task is to predict the echo top height. This, of course, is dependent on a $45\,\mathrm{dBZ}$ reflectivity being present in the vertical column. Thus, we divide this task into two components: predicting whether a $45\,\mathrm{dBZ}$ echo top will be present (ECHO45-OCC) and, in cases where it is present,

predicting its height (ECHO45-HT). In an operational setting, these model could be used by first evaluating ECHO45-OCC; if it predicted that a 45 dBZ reflectivity would occur, ECHO45-HT would then be predicted and the freezing level would be subtracted from it in order to calculate the hail probability.

### 3.3 Machine learning: Gradient tree boosting

For the ML prediction, we use gradient boosting (GB) to learn the relationship between the features and the prediction targets.
GB is a ML technique that uses decision trees, training trees iteratively such that each successive tree corrects the errors of the previous trees. The decision trees are regularized using several techniques in order to prevent overfitting. A review of GB methods can be found in Natekin and Knoll (2013).

One particular advantage of GB for our study is that it allows the importance of the various input features to be quantified. Thus, it suits well our purpose of assessing the value of different data sources and variables for the prediction of thunderstorm
hazards. The results can be used to later guide the selection of appropriate features for different ML methods such as deep learning where the feature importance is less straightforward to derive.

We used the open-source LightGBM implementation of the GB algorithm (Ke et al., 2017) as our ML framework. LightGBM is designed to be computationally efficient and has a reduced memory footprint, facilitating the analysis of large datasets.

### 3.4 Training

We tuned the GB training for each of the various prediction tasks. As described in Sect. 3.2, the learning tasks can be broadly divided into two categories: regression tasks and binary classification tasks. In regression tasks, the objective is to predict the future value of a variable that can be any real number, while in binary classification tasks the objective is to predict the probability of an event occurring.

After comparing the performance and robustness of mean square error (MSE) and mean absolute error (MAE), we decided
to use MAE as the training objective function for regression tasks as it tended to give slightly better results with the validation set, with less overfitting. Indeed, when the model was trained with MAE loss, it achieved better MSE in the validation set than an equivalent model trained with MSE loss. Willmott and Matsuura (2005) and Chai and Draxler (2014), among others, have discussed the relative merits of MAE and MSE in the geoscientific context. We also found that, compared to training the

GB model for the target variable directly, we can achieve superior training performance by first subtracting the bias-corrected persistence prediction (discussed in more detail in Sect. 4.1) and then training the GB model to predict the residual. Binary tasks are trained using the cross entropy as a cost function.

All tasks are trained using early stopping based on the validation dataset. That is, the training proceeds as long as the training metric keeps improving not only in the training set, but also in the validation set which is not used for training. The early stop limits the overfitting of the GB model.

The performance with the validation dataset was also used to tune the hyperparameters of the GB model, most importantly the depth of the trees, the number of leaf nodes, the learning rate and various regularization parameters. Although we were able to achieve some improvements by fine-tuning these parameters, the performance on the validation set was not particularly sensitive to changes over a reasonable range of parameters. As the principal goal of this study is not to strictly optimize the performance of the predictions but rather to assess the importance of the various data sources, we consider the hyperparameter tuning to be of secondary importance in this context, and are content to use hyperparameters that produce reasonable results after an informal manual search of the parameter space.

Using the default hyperparameters, approximately $60 \, \mathrm{min}$ was required on a modern computer with 16 central processing unit (CPU) cores to train the all GB models: 12 models corresponding to different time steps of the MAXZ prediction and 2 models (0–30 min and 30–60 min) for each of the LIGHTNING-OCC, ECHO45-OCC and ECHO-45. Thus, one model took approximately $3 \, \mathrm{min}$ to train. Evaluating all of the above-mentioned models for the entire testing dataset of 8872 samples required a total of $6 \, \mathrm{s}$ on the same hardware; this is equivalent to $35 \, \mu\mathrm{s}$ for one sample and one model.

## 4    Results and discussion

The results of the ML experiments are reported and discussed in the sections below. First, in Sect. 4.1 we give a general analysis of the prediction performance. Then, we assess the importance of different features and data sources using the GB feature importance (Sect. 4.2) and data exclusion analysis (Sect. 4.3). All reported results are for the test dataset unless otherwise mentioned.

### 4.1    Prediction performance

Before evaluating the importance of the various data sources, we quantify the performance of the models in the case where all data sources described in Sect. 2.2 are available.

Figure 3 shows examples of the real and predicted time series for MAXZ. We note that in this figure, the MAXZ at $t = 0$ may be less than the 37 dBZ threshold because the MAXZ shown is the mean over the 25-km diameter region of interest, while MAXZ exceeding 37 dBZ in a single pixel is enough for a case to be selected. Meanwhile, Fig. 4 shows the error, averaged over all events and data points, of the MAXZ predictand as a function of the lead time $t$. In order to provide a more concrete error figure, we also show the corresponding relative error in the rain rate estimated using the relation $Z = 300R^{1.4}$, where $Z$ is the reflectivity on the linear scale, derived for convective precipitation and frequently used with NEXRAD (e.g. Martner

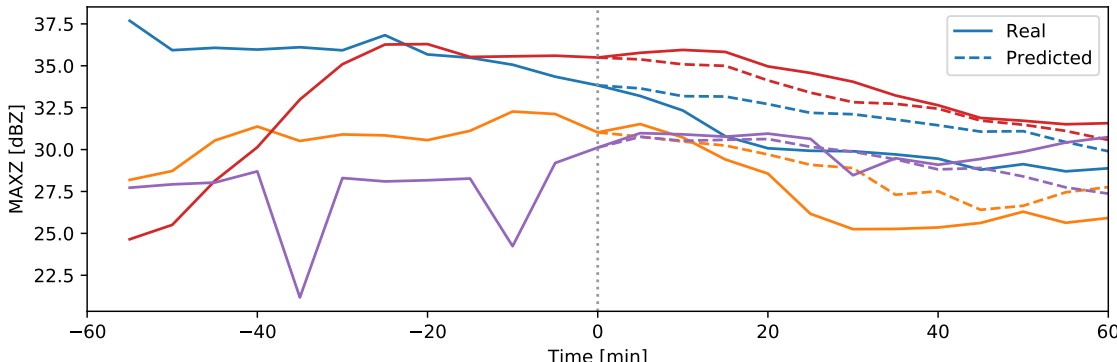

**Figure 3.** Examples of the prediction of MAXZ. The figure show the curves of observed and predicted MAXZ (the mean over the 25-km diameter region of interest) for four different tracked centers. The solid lines show the development of MAXZ while the dashed lines show the predictions after $t = 0$.

et al., 2008). As a baseline prediction, we use the persistence assumption in a Lagrangian framework, that is, it is assumed that the variable will remain the same as it was at time $t = 0$. We found that the persistence assumption is biased: the MAXZ at $t > 0$ is, on average, lower than at $t = 0$; this can also be seen in most of the examples of Fig. 3. This reflectivity bias is caused by a combination of two sources: first, sampling bias which occurs because we select centers of intensive thunderstorms with
310 $\mathrm{MAXZ} > 37$ dBZ, and second, the thunderstorm track drifting off the actual center of the storm due to inaccuracies in the tracking procedure. The bias is small at short lead times and reaches $3.5$ dBZ at $t = 60$ min.

We can considerably reduce the error of the persistence assumption by correcting for this bias. In contrast, it is rather difficult to improve from the bias-corrected persistence assumption using the GB model, even if we train the GB model on its residual. In Fig. 4, it is apparent that the bias correction improves the MAE by approximately $1.2$ dB at $t = 60$ min while the GB model
only gives a further $0.3$ dB of improvement. Nevertheless, the improvement gained with the ML prediction is consistent and increases with longer lead times.

For the lightning prediction, the model has an error rate of $8.1\%$ for LIGHTNING-OCC in the $0$–$30$ min time period and $14.3\%$ for the $30$–$60$ min period. We can compare these numbers to the climatological occurrence, which would be the error rate of a prediction that lightning never occurs — or conversely, the climatological non-occurrence would be the error
rate of a prediction that it always occurs. The climatological occurrence of lightning in the data is $40.7\%$ for $0$–$30$ min and $29.2\%$ for $30$–$60$ min in the test dataset (the difference in these is likely due to the same bias that we discussed in the context of MAXZ above). These results suggest that the nowcasting framework developed here could potentially be adapted to operational lightning nowcasting. Full confusion matrices are shown in Fig. 5a–b.

For the presence of the $45$ dBZ echo (ECHO45-OCC), we find error rates of $12.7\%$ for the $0$–$30$ min range and $16.5\%$ for
the $30$–$60$ min range. The corresponding climatological occurrences in the test set are $38.3\%$ for $0$–$30$ min and $20.0\%$ for the $30$–$60$ min range. Thus, we achieve a considerable improvement with the ML approach for the near-term prediction but a far

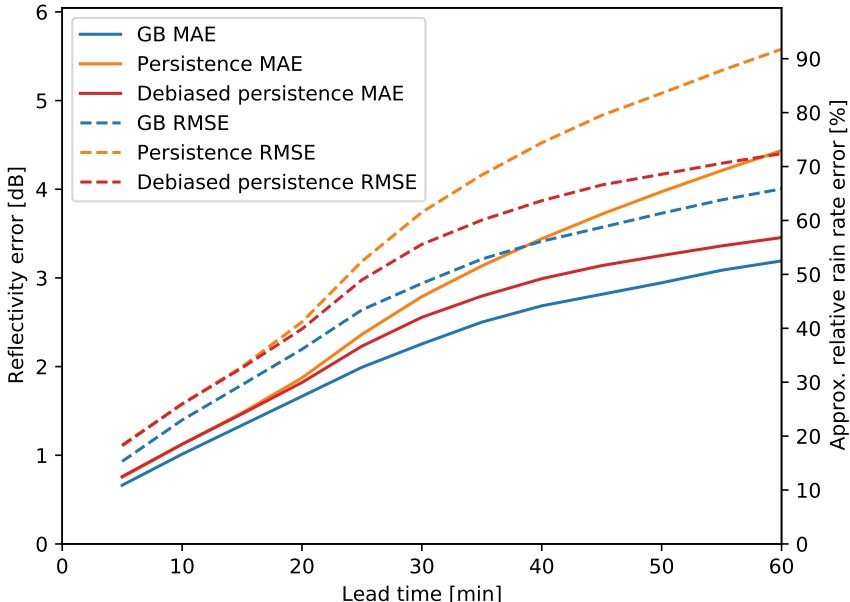

**Figure 4.** Errors in dBZ of the prediction of MAXZ as a function of lead time, using all the available data. The solid lines show the mean absolute error (MAE) while the dashed lines show the root-mean-square error (RMSE) as shown on the scale on the left. The scale on the right shows the logarithmic reflectivity error converted to the relative error in rain rate $R$. The blue lines show the result from the GB tree, the orange lines show the Lagrangian persistence assumption and the red lines show the bias-corrected Lagrangian persistence.

more marginal one for the longer term, which implies a more limited ability to predict hail, and other features associated with the 45 dBZ echo top, using the approach used in this study at lead times over 30 min. The confusion matrices for ECHO45-OCC can be found in Fig. 5c–d. In the subset of the test dataset where the 45 dBZ echo is present, the height of the 45 dBZ echo is predicted with a MAE of 693 m for 0–30 min and 841 m for 30–60 min. According to the formula of Foote et al. (2005) for the probability of hail (POH), these correspond to roughly 16 and 19 percentage point errors in POH, respectively. Meanwhile, the standard deviations of ECHO45-HT in the test set are 1365 m for 0–30 min and 1404 m for 30–60 min.

## 4.2 Feature importance

The importance of features to the GB model can be extracted from the LightGBM library after training. In this section, we show the "gain" of various features, i.e. the total reduction of the training loss function attributable to that feature. For clarity and brevity of presentation, we sum together the contributions from different feature types (e.g. mean, standard deviation) and different past time steps. We also separately consider the total contribution from all features of a given data source, which can give a clearer impression of the total importance of a source that includes a large number of correlated features.

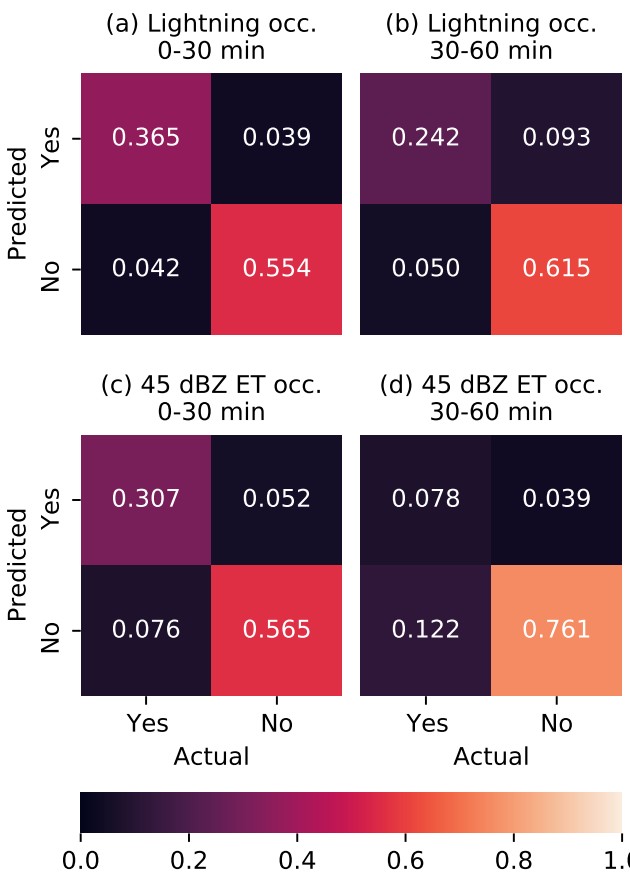

**Figure 5.** Confusion matrices for (a–b) the LIGHTNING-OCC prediction task and (c–d) the ECHO45-OCC task.

Figures 6 and 7 show the importance of the various predictors and data sources (e.g. ABI, NWP model or radar) for each of the ML objectives defined in Sect. 3.2. In Fig. 6, we show the feature and source importances of the MAXZ and LIGHTNING-OCC objectives, while Fig. 7 displays the same for ECHO45-OCC and ECHO45-HT.

The statistics of feature importance for nowcasting MAXZ, in Fig. 6a, demonstrate that the most important features for predicting this target variable come from the NEXRAD radar data. The most significant feature is the column maximum reflectivity, the same variable that is being predicted, but the other radar variables also seem to be utilized. The importances grouped by data source, displayed in Fig. 6b, show a slightly different view, as there the NWP data are of similar importance compared to the radar. The reason for the apparent discrepancy between the importances of individual features and the total source importance is that contribution of the NWP data is divided over a large number of variables that are correlated to varying degrees. Because of the correlations, the contributions from individual variables appear small as the GB model splits the gain between the correlated variables, but the contributions combine to an importance comparable to the radar variables when summed together. Figure 6b shows some variation in the relative importance between the radar and NWP data between time steps, which

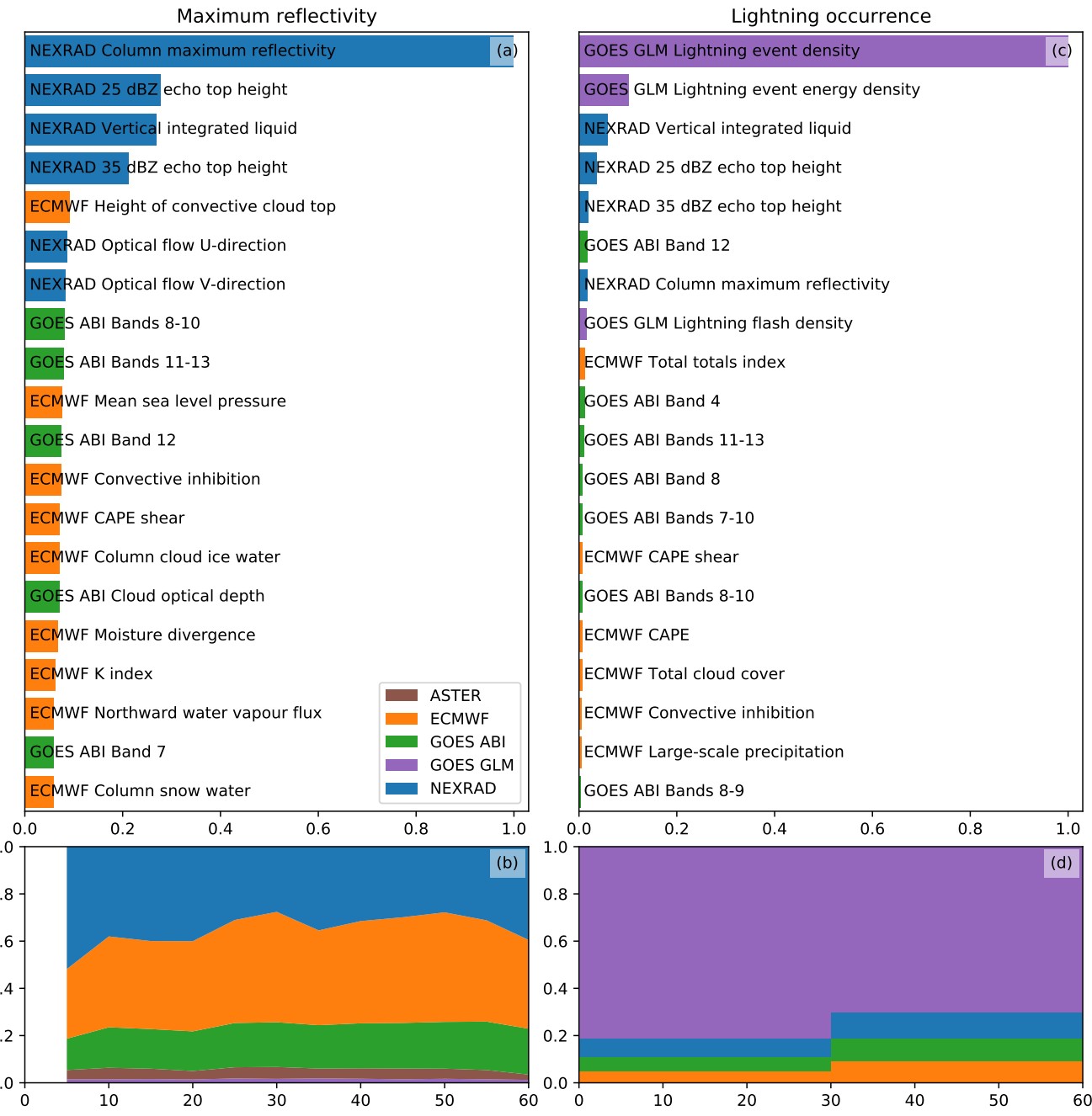

**Figure 6.** The importance of the various features and data sources for the MAXZ (panels a and b) and LIGHTNING-OCC (panels c and d) predictands according to LightGBM. The top panels show the 20 most important source variables for each predictand. The importances have been summed together from all features and time steps and normalized such that the most important variable is scaled to 1. The bottom panels show the total importance of the various data sources as a function of lead time (in panel b) or the prediction time range (in panel d).

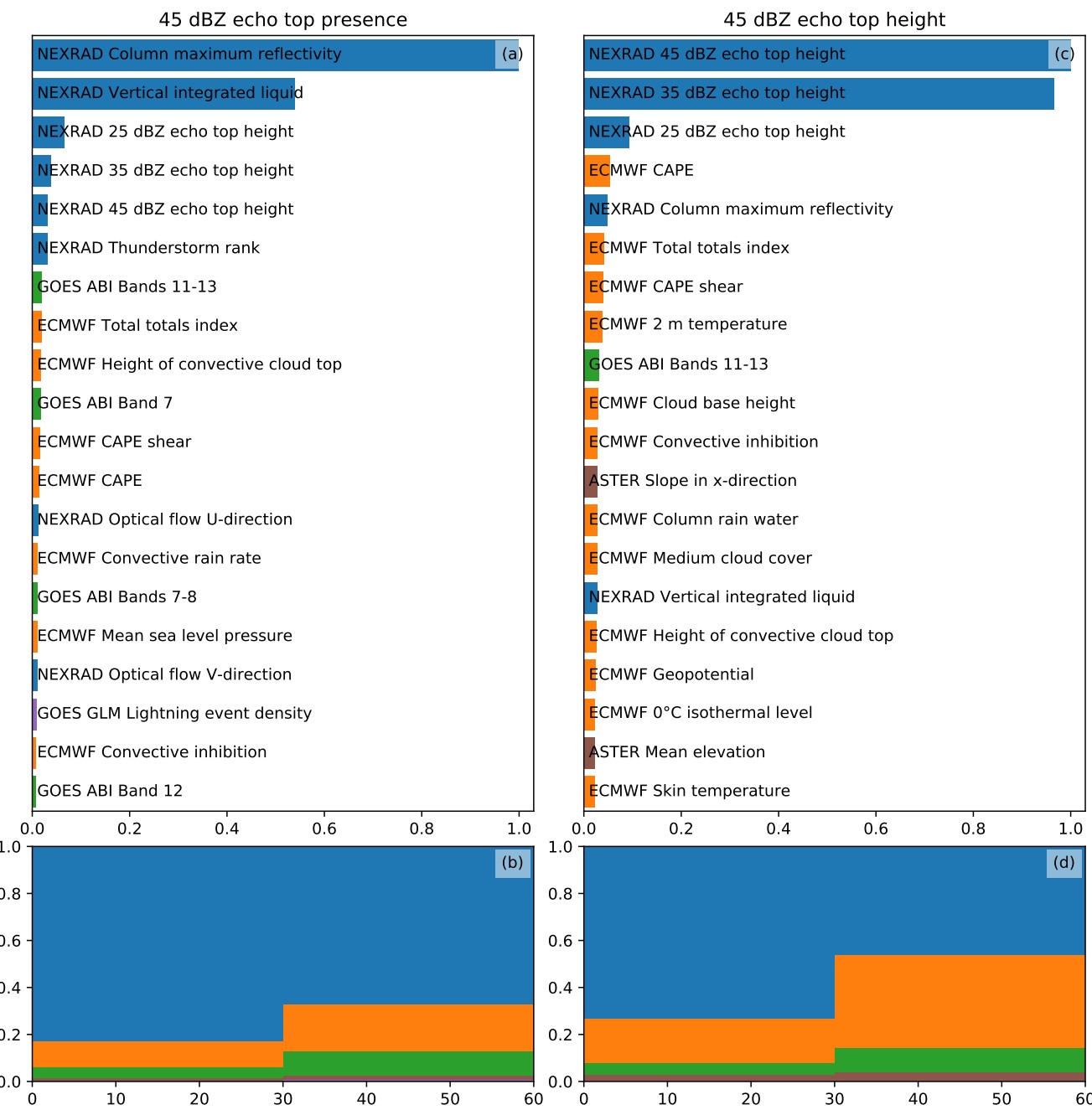

**Figure 7.** As Fig 6, but for the ECHO45-OCC (panels a and b) and ECHO45-HT (panels c and d) predictands.

we consider to be most likely mere random noise; as the different time steps are predicted by different, independently trained models, they may end up with slightly different values for the feature importance. In general, the importance of the NWP data tends to increase slightly with longer lead times, as was also found by earlier nowcasting studies (e.g. Kober et al., 2012). The GOES-16 ABI data are also utilized to a significant degree, while the GLM and ASTER data contribute to a lesser extent.

The feature and source importances for LIGHTNING-OCC, shown in Fig. 6c–d, are dominated by contributions of the GLM lightning data. This is largely because a region that is already producing lightning is likely to continue to do so in the future, thus providing a reliable predictor, but past occurrence can also indicate temporal tendencies in lightning activity. The NEXRAD, ABI and ECMWF data are considered approximately equally important, with their total importance (Fig. 6d) relative to GLM increasing with longer lead times.

Figure 7 shows the feature importances for ECHO45-OCC and ECHO45-HT. Similar to the nowcasting of MAXZ, the most important features are from the NEXRAD radar data. Again, this is not unexpected given that the target variables are defined using the radar. The ECMWF and ABI data contribute less to the prediction of the echo top than to the prediction of MAXZ. According to this analysis, the GLM data are hardly used, while the ASTER DEM data seems to provide a small contribution to the prediction of echo top height. However, as we shall discuss in Sect. 4.3, GLM actually provides useful data in the absence of other predictors, while the importance of the DEM may be due to overfitting.

## 4.3 Exclusion studies

An alternative way to assess the importance of various data sources is to remove one or more data sources from the training data, retrain the model, and evaluate the change in prediction performance. This approach may give later studies a clearer picture of the value of various data sources in thunderstorm nowcasting applications. Unlike the feature importance, such an exclusion study also allows the use of the testing set for evaluation, showing which variables are important in practice and allowing us to better distinguish generalizing learning ability from overfitting. The results of the exclusion experiments are shown in Figs. 8 (MAXZ and LIGHTNING-OCC) and 9 (ECHO45-OCC and ECHO45-HT). We also show the equivalent results for the training and validation datasets in the Appendix Figs. A1–A4.

The results for the MAXZ predictand at $60$ min lead time can be found in Fig. 8a–b. There is some noise in the results, so small differences in the metrics should not be overinterpreted, but certain general patterns are apparent. Most noticeably, the two leftmost columns, which correspond to models that have the NEXRAD radar data available, show consistently lower errors than the two columns on the right. The ABI data also have a positive effect, as can be seen by comparing the first column to the second, or the third to the fourth. Examining the differences between the rows, GLM data have a slight positive effect especially when few other data sources are available, while it is difficult to discern any consistent effect from including the ECMWF data, and including the ASTER data sometimes even appears to make the metrics slightly worse. The latter result may be caused by the GB training process overfitting to the DEM features during training, degrading the results obtained during testing. Among the predictions obtained using only one data source, the one with NEXRAD data yields the best results and is almost as good as using all data sources together, the ABI and GLM data provide slight improvements over the baseline case (shown in the bottom right corner), while the model using only the NWP data from ECMWF yields results approximately equal

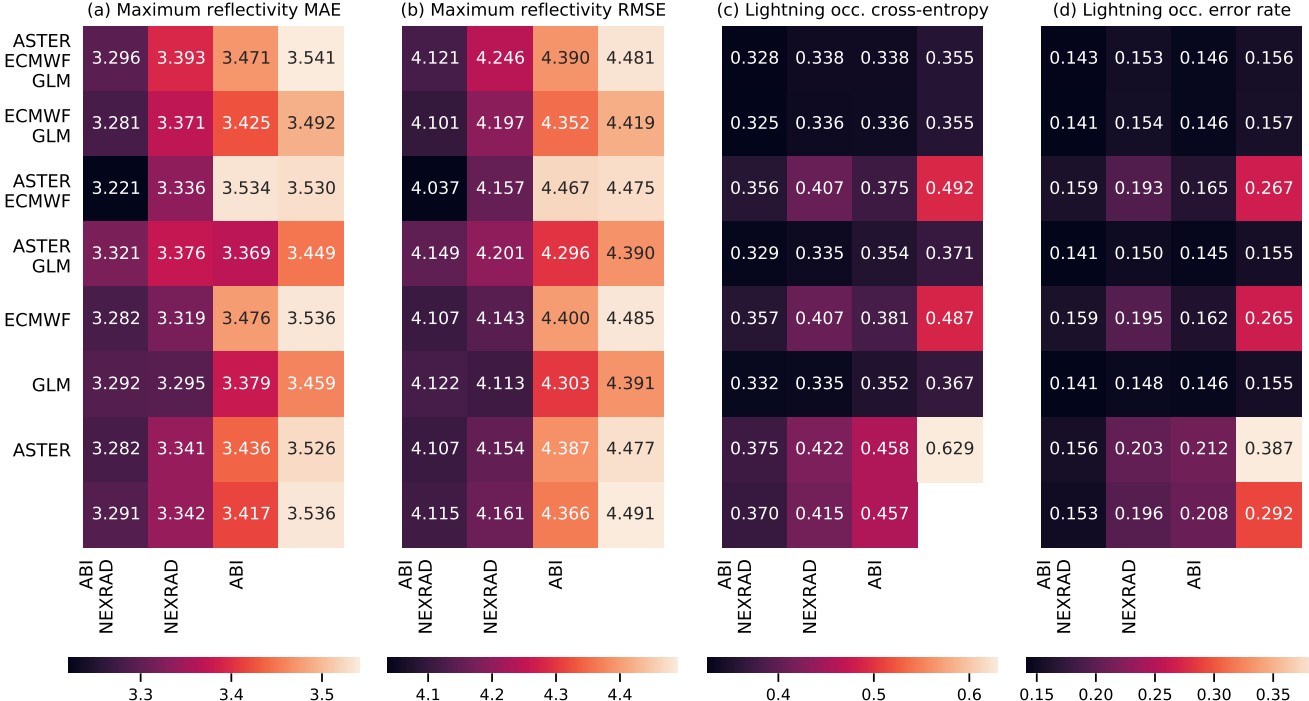

**Figure 8.** The results of exclusion experiments on the MAXZ and LIGHTNING-OCC predictands. Each square in the panels a–d corresponds to a combination of data sources, which can be found by combining sources listed for the row and the column. For example, the top left square of each panel shows the error metric obtained using all five data sources, while the second column of the second row shows the metric for ECMWF, GLM and NEXRAD data. The predictand and the error are shown on top of each panel; RMSE indicates the root-mean-square error and MAE the mean absolute error. In panels a and b, the bottom right corner shows the result obtained with the bias-corrected persistence assumption (see Sect. 4.1), while in panel d, the bottom right corner shows the baseline climatological occurrence. The results for LIGHTNING-OCC are shown for the 30–60 min time interval.

to the persistence baseline. The latter result is quite surprising considering the large weight assigned to the ECMWF features in the feature importance analysis (Fig. 6a–b). For unclear reasons, the single best combination seems to be that which uses all data sources except GLM; we suspect that this result is merely coincidental and due to random variation because the results in Fig. 8a–b do not suggest that the GLM data is detrimental to prediction performance. The results for the training and validation sets (Appendix Figs. A1a–b and A3a–b) support this interpretation since in the validation dataset the best combination is that of NEXRAD and GLM. More generally, the results for the training and validation datasets exhibit patterns similar to those in the test set, which suggests that while individual differences may be attributable to noise, the broader conclusions of the analysis are robust.

The metrics for LIGHTNING-OCC, shown in Fig. 8c–d for the 30–60 min time interval, also show a clear pattern. Here, the first, second, fourth and sixth row, which correspond to the GLM data being available, show better metrics than the other rows.

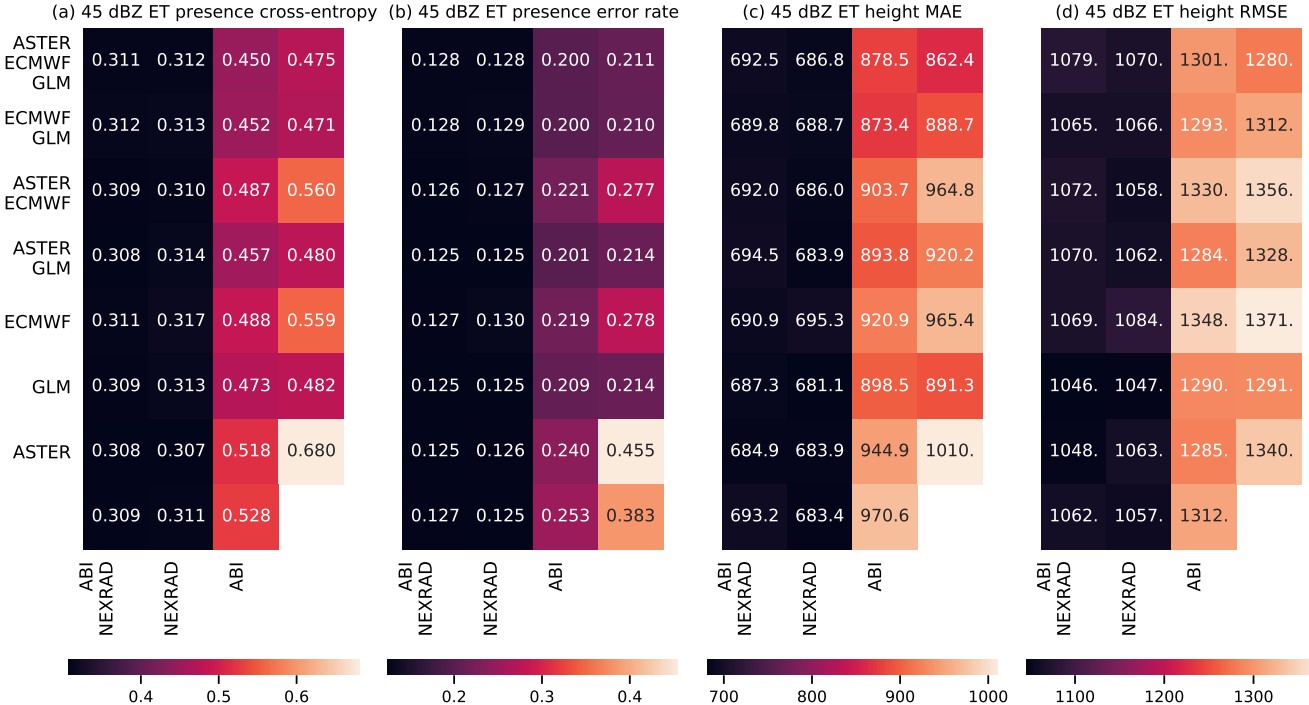

**Figure 9.** As Fig. 8, but for the ECHO45-OCC and ECHO45-HT predictands. In panel b, the bottom right corner shows the climatological occurrence (unlike with MAXZ, we do not use the persistence assumption as a baseline for ECHO45-HT, and therefore nothing is shown in the bottom right corner of panels c and d in contrast to Fig. 8a–b). The results are shown for the 0–30 min time interval.

Indeed, prediction using *only* the GLM data performs very well, achieving an error rate of 15.5%. However, it is interesting to note that good results can be obtained without the direct lightning data as well: For example, the error rate obtained using only the ABI and NEXRAD (15.3%) data is better than the GLM-only error rate, and only 1.2 percentage points worse than the best result (14.1%). This shows that the feature importance analysis shown in Sect. 4.2 is only valid for a specific combination of predictors. When one data source is removed, the missing information can often be substituted by other data sources that were much less used in the case in which everything was available. In general, both the ABI and NEXRAD data improve the prediction results for LIGHTNING-OCC: The first column (with both ABI and NEXRAD available) has the best results overall, the comparison between the second (NEXRAD only) and the third (ABI only) is mixed, and the fourth (neither ABI nor NEXRAD available) has the worst results. The results with the ECMWF data are rather odd: First, in contrast to the MAXZ prediction, the ECMWF data has some skill on its own in lightning prediction, as the ECMWF-only prediction has an error rate 2.7 percentage points better than the climatological average of 29.2%. Second, the prediction with both ECMWF and ABI performs 4.6 percentage points better than the ABI prediction alone, but meanwhile adding the ECMWF data to NEXRAD does not offer an improvement over the NEXRAD-only scores; this result may be simply noise as in the validation dataset (Appendix Fig. A3d) the addition of the ECMWF data also improves the results with NEXRAD. The ASTER data does not

add much information, and the ASTER-only prediction with 38.7% error rate actually performs worse than the climatology. This happens because the climatological occurrence in the training dataset, at 42.5%, is coincidentally significantly higher than in the test dataset. The ASTER-only model is unable to generalize with the scarce data available to it, and only learns to roughly reproduce the climatological error rate in the training dataset, which leads to overestimation of occurrence in the test set. Indeed, the degradation of the metrics with the addition of ASTER does not occur in the test and validation datasets.

For both ECHO45-OCC and ECHO45-HT (Fig. 9, shown for the 0–30 min interval), the clearest pattern is the importance of the radar data for prediction, consistently with the feature importance analysis. Indeed, as long as the NEXRAD data are available, the benefit of adding further data sources is negligible compared to the NEXRAD-only case (error rate of 12.5%). However, without the NEXRAD data (e.g. in oceanic regions without radar coverage) the other data sources still provide meaningful improvements in ECHO45-OCC over the climatological occurrence of 38.3%. For example, ABI alone achieves an error rate of 25.3%, ECMWF alone yields 27.8%, and GLM alone achieves 21.4%, while these three sources together reach 20.0%. Similar patterns are found in ECHO45-HT. These results further demonstrate that the benefits of features and data sources cannot be evaluated in isolation and depend on what other data sources are used.

## 5  Conclusions

For machine-learning methods to be utilized effectively for thunderstorm nowcasting, it is necessary that the benefits of the various available data sources be well understood and quantified. Large amounts of data that are potentially related to convective processes can be obtained from numerous sources, yet it is not always obvious how much benefit one should expect from adding an additional data source, and therefore additional complexity, to a ML model. This study provides guidance for future work to better select data sources for nowcasting particular thunderstorm hazards along predicted storm tracks. We obtained data from ground-based radar (NEXRAD), satellite spectrographic imagery (GOES-16 ABI), satellite-based lightning detection (GOES-16 GLM), a numerical weather prediction model (ECMWF IFS) and a digital elevation model (ASTER), for a total of over 100 input variables. We applied this data to nowcast variables related to precipitation, lightning and hail formation.

We have based our evaluation of the importance of various features on two complementary approaches: first, using the feature importance provided by the gradient boosted tree algorithm, and second, retraining the GB algorithm repeatedly using different subsets of the input variables. Testing all possible combinations of input features would have quickly become implausible as the number of features increased, but grouping the features by data source allowed us to cover the most realistic situations of missing data, where an entire data source is unavailable due to either geographical limitations (for example, operational weather radar networks do not cover the oceans) or irregular data outages.

Each of the investigated data sources proved to be useful for predicting at least some of the target variables, except for the DEM that provides no detectable benefit for any of the predictands. The radar variables are strong predictors for all predictands, and are particularly dominant for the targets defined using the radar data. The satellite imagery from ABI provides moderate performance improvements to all predictands, though it is generally less significant than the radar data in this application. The GLM lightning data are highly useful for lightning prediction; for other targets, they provide more modest benefits, although

they can still provide improvements to nowcasting performance particularly when radar data are not available. More generally, the results confirm that satellite data can be used to provide ML-based nowcasts in areas without radar coverage, such as over the oceans and in less-developed regions lacking ground-based radar networks. Meanwhile, the ECMWF forecast data, despite being considered of some importance by the ML algorithm, do not benefit the nowcast according to the data exclusion analysis, as for the lead times investigated here, the necessary information content is already contained in the other observations.

The results show that the feature importance from the GB algorithms may provide seemingly contradictory results compared to the more comprehensive analysis achieved by testing different combinations of features and evaluating the results. Although the two evaluation methods largely agreed on which data sources are the most important, some important differences emerged on closer inspection. This highlights the pitfalls of analyzing the importance of features and data sources in an ML setting when the data sources are partially redundant. A given feature may be beneficial when used alone, but virtually useless when used in conjunction with another, more powerful predictor. For instance, when trained for lightning prediction, the ML algorithm only gains a modest improvement (approximately $10\%$) in the error rate from auxiliary data sources when direct lightning data are available, but when trained without lightning data, good prediction performance can still be achieved utilizing the other data sources. This has important implications for real-time nowcasting in time-critical applications such as aviation, as it indicates that ML-based nowcasting can be performed robustly when some input data are missing or delayed.

Based on the results, we conclude that investigators should be cautious with applying the brute-force strategy of providing ML algorithms with all the available data and letting the training process decide which data are useful. While this may sometimes reveal unexpectedly useful input variables, using data sources that contain little or no generalizable information may also expose the training process to the problem of overfitting, thus actually degrading the results. This can be mitigated by early stopping and by using hyperparameters designed to prevent overfitting, but it is better for both accuracy and training time to simply drop the counterproductive data sources.

Future work can take advantage of the results achieved in this study to build more accurate and efficient ML models for the nowcasting of thunderstorm hazards of heavy precipitation, lightning and hail. It also allows the estimation of the degradation of the result if one observation system is missing. Nevertheless, this work is constrained to a particular set of data sources, a single study area and a specific ML method: the gradient boosted tree. Although we have selected five commonly utilized data sources, all qualitatively different from each other, later work should extend the analysis to other data sources such as polar-orbiting satellites and ground-based lightning networks. Given suitable data sources, the methodology could also be extended to other hazards like wind damage and tornado events. It may furthermore be interesting to investigate additional regions: For instance, the DEM data may be more significant in regions with higher mountains. With regard to alternative ML methods, the performance of neural networks should be evaluated in a future study, preferably using the same dataset to facilitate comparisons, as convolutional neural networks are expected to be able to better utilize spatial features such as the high-resolution imagery from the ABI instrument. Neural networks may also be able to utilize large numbers of samples and input variables better.

*Code and data availability.* The ML code used to produce the results is available at https://github.com/meteoswiss-mdr/ts-nowcast-datasources. The feature dataset used to train the ML models can be found at Leinonen et al. (2021).

The original datasets are described, with instructions for downloading and reading, in the following sources in the References:

– NEXRAD radar data: NOAA National Weather Service (NWS) Radar Operations Center (1991)

– GOES ABI L1b data: GOES-R Calibration Working Group and GOES-R Series Program (2017)

– GOES ABI L2 products:

    – Cloud top height: GOES-R Algorithm Working Group and GOES-R Series Program Office (2018a)

    – Cloud optical depth: GOES-R Algorithm Working Group and GOES-R Series Program Office (2018b)

    – Cloud top pressure: GOES-R Algorithm Working Group and GOES-R Series Program Office (2018c)

    – Derived stability indices: GOES-R Algorithm Working Group and GOES-R Series Program Office (2018d)

– ASTER GDEM Version 3: NASA/METI/AIST/Japan Spacesystems and U.S./Japan ASTER Science Team (2019)

The ECMWF forecast archive is available only to licensed users and participating national meteorological services; these can obtain the data through the ECMWF Meteorological Archival and Retrieval System (MARS).

**Appendix A: Exclusion studies on training and validation datasets**

We also performed the exclusion analyses, shown in Sect. 4.3 for the test dataset, with the training and validation datasets. The
490 results are shown in Figs. A1–A2 for the training set and in Figs. A3–A4 for the validation set.

**Appendix B: Further information on the datasets**

Table A1 lists the radars used to compile the NEXRAD dataset we used. Tables A2–A6 list the predictors from the various data sources that were used in this study.

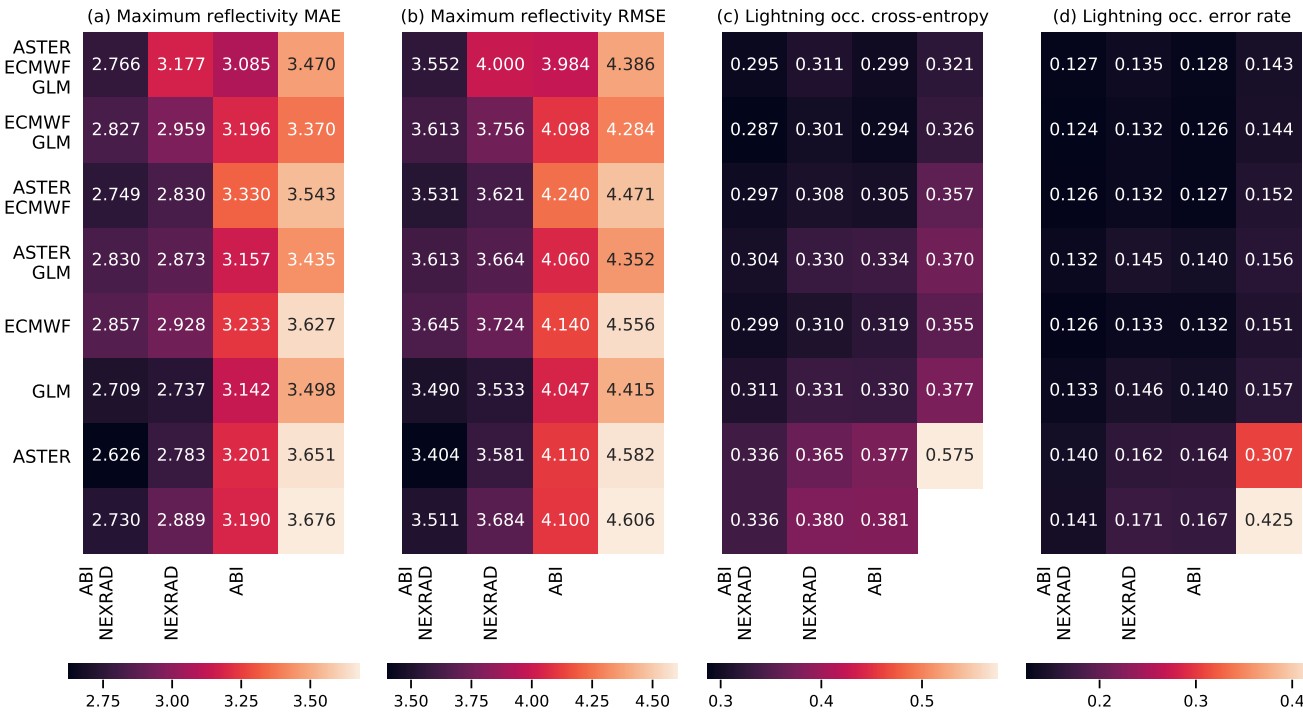

**Figure A1.** As Fig. 8, but for the training dataset.

**Table A1.** NEXRAD radars used to produce the dataset used in this study.

| Location | Code |
|---|---|
| Albany, New York | KENX |
| Binghamton, New York | KBGM |
| Buffalo, New York | KBUF |
| Burlington, Vermont | KCXX |
| Boston, Massachusetts | KBOX |
| Fort Drum, New York | KTYX |
| New York City, New York | KOXZ |
| Philadelphia, Pennsylvania | KDIX |
| Portland, Maine | KGYX |
| Pittsburgh, Pennsylvania | KPBZ |
| State College, Pennsylvania | KCCX |

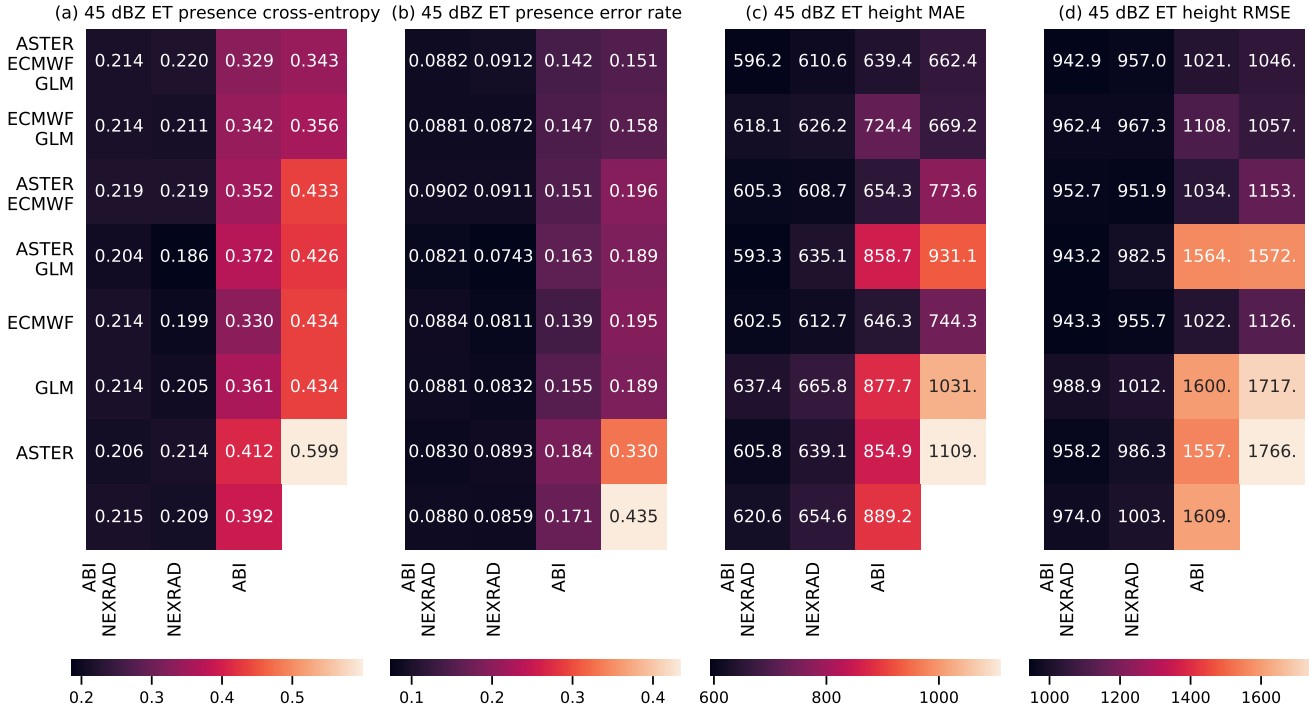

**Figure A2.** As Fig. 9, but for the training dataset.

**Table A2.** Variables from the NEXRAD radar adopted in this study.

| |
|---|
| 25 dBZ echo top height |
| 35 dBZ echo top height |
| 45 dBZ echo top height |
| Maximum reflectivity |
| Vertically integrated liquid |
| Motion U/V components from from optical flow |

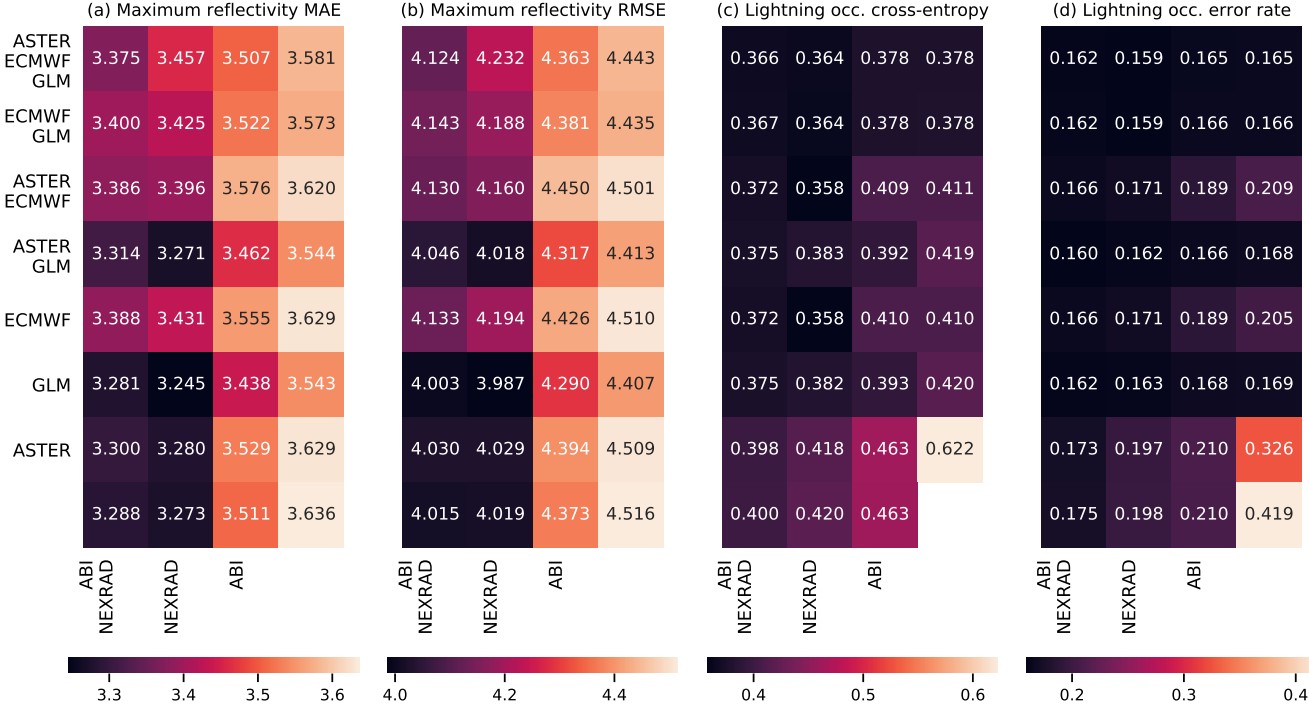

**Figure A3.** As Fig. 8, but for the validation dataset.

**Table A3.** Variables from the GOES-16 ABI instrument adopted in this study.

| Level 1 | | | Level 2 |
|---|---|---|---|
| ABI band 01 (0.47 μm) | ABI band 09 (6.9 μm) | Difference 07-08 | Cloud top height |
| ABI band 02 (0.64 μm) | ABI band 10 (7.3 μm) | Difference 07-09 | Cloud top pressure |
| ABI band 03 (0.86 μm) | ABI band 11 (8.4 μm) | Difference 07-10 | Cloud optical depth |
| ABI band 04 (1.37 μm) | ABI band 12 (9.6 μm) | Difference 08-09 | CAPE |
| ABI band 05 (1.6 μm) | ABI band 13 (10.3 μm) | Difference 08-10 | K-index |
| ABI band 06 (2.2 μm) | ABI band 14 (11.2 μm) | Difference 11-13 | Lifted index |
| ABI band 07 (3.9 μm) | ABI band 15 (12.3 μm) | Difference 12-13 | Showalter index |
| ABI band 08 (6.2 μm) | ABI band 16 (13.3 μm) | | Total totals index |

**Table A4.** Variables from the GOES-16 GLM instrument adopted in this study.

| |
|---|
| Flash density |
| Flash energy density |
| Event density |
| Event energy density |

**Table A5.** Variables from the ECMWF model output adopted in this study.

| | |
|---|---|
| 0°C isothermal level | Mean sea level pressure |
| 10 m $U/V$ wind components | Medium cloud cover |
| 100 m $U/V$ wind components | Potential evaporation |
| 200 m $U/V$ wind components | Precipitation type |
| 2 m dewpoint temperature | Skin reservoir content |
| 2 m temperature | Skin temperature |
| Boundary layer dissipation | Snowfall |
| Boundary layer height | Surface latent heat flux |
| Cloud base height | Surface pressure |
| Convective available potential energy | Surface net solar radiation |
| Convective available potential energy shear | Surface net solar radiation, clear sky |
| Convective inhibition | Surface net thermal radiation |
| Convective precipitation | Surface net thermal radiation, clear sky |
| Convective rain rate | Surface sensible heat flux |
| Convective snowfall rate water equivalent | Total cloud cover |
| Evaporation | Total column cloud ice water |
| Friction velocity | Total column cloud liquid water |
| Geopotential | Total column rain water |
| Height of convective cloud top | Total column snow water |
| Height of 1°C wet-bulb temperature | Total column supercooled liquid water |
| Height of 0°C wet-bulb temperature | Total column water |
| High cloud cover | Total column water vapour |
| $K$ index | Total precipitation |
| Large-scale precipitation | Total precipitation rate |
| Large-scale precipitation fraction | Total totals index |
| Large scale rain rate | Vertically integrated moisture divergence |
| Large scale snowfall rate water equivalent | Vertical integral of eastward water vapour flux |
| Low cloud cover | Vertical integral of northward water vapour flux |

**Table A6.** Variables from the ASTER DEM adopted in this study.

| |
|---|
| Mean elevation |
| Roughness |
| Surface gradient |
| Upslope flow |

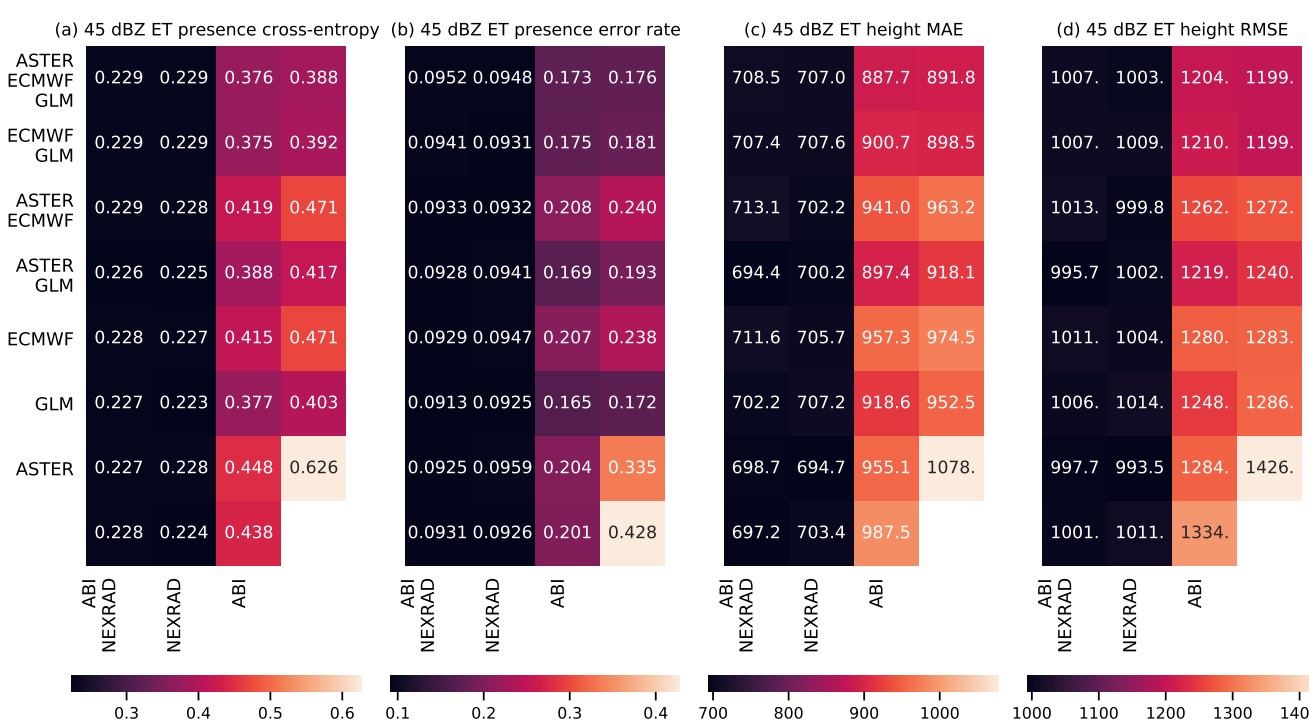

**Figure A4.** As Fig. 9, but for the validation dataset.

*Author contributions.* UH, UG and JM conceived the initial concept of the study, which was refined by JL and UH. With the support of UH, JL obtained the data, processed them, trained the ML models and analysed the results. JL led the writing of the manuscript, with contributions from all co-authors.

*Competing interests.* The authors declare that they have no competing interests.

*Financial support.* The work of JL was supported by the fellowship "Seamless Artificially Intelligent Thunderstorm Nowcasts" from the European Organisation for the Exploitation of Meteorological Satellites (EUMETSAT). The hosting institution of this fellowship is MeteoSwiss in Switzerland.

*Acknowledgements.* This study builds on the work realized in the COALITION-3 project of MeteoSwiss, to which Joel Zeder and Shruti Nath contributed significantly. We thank David Haliczer and Christopher Tracy from the University of Alabama in Huntsville for their support in defining the study area and working with the US datasets, and Lorenzo Clementi of MeteoSwiss for comments on the initial draft of the article. We also thank Tomeu Rigo and two anonymous reviewers for their constructive feedback on this article.

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
