# Peer review of "Nowcasting thunderstorm hazards using machine learning: the impact of data sources on performance"

_Natural Hazards and Earth System Sciences, 2021_

## Referee Comment (RC1)

Title: **Nowcasting thunderstorm hazards using machine learning: the impact of data sources on performance**

Authors:  **Jussi Leinonen, Ulrich Hamann, Urs Germann, and John R. Mecikalski**

Journal: **NHESS**

General comments

Nowcasting techniques based mainly on radar data have suffered in the last two decades a stationarity in their development, caused by different factors. One of them is the high degree of confidence of the classic procedures, except in some complicated cases. Besides, the difficulty for finding a technique able to remove or, at least, reduce the gap between the nowcasted and the observed results in the extreme cases has produced this stop in this field. However, during this period, many new proposals have been developed and presented, considering mainly statistics and historical data. Machine learning is one of these techniques, and probably, one of those that produce largest expectations in the early future.

The present manuscript analyzes the use of the machine learning from an operational point of view, which results absolutely necessary in the sense that considering the increase of severity of thunderstorms in the last and future decades, meteorologists should provide better warnings to population. Because some of the data types are not still available in Europe, the authors have considered a region with similar geographic conditions, where all the data set are accessible.

The document is well-addressed, easy to follow, and well-documented. My major concerns regard on the type of data and/or methodologies, according to the main objective: "we seek to understand the impact on thunderstorm nowcasting from the new generation of geostationary satellites, which, compared to the previous generation, provide higher-resolution imagery, additional image channels and lightning data ":

- In my opinion, there is a lack of coherence on the considered model: if the authors analyze an American region and all the data are from American sources, why in the case of the NWP they have considered the European one. Could you clarify this point?
- In a similar way, if the analysis is focused on operational methods, why you do not have taken into account the operational methodology (or a similar one, maybe provided by the NWS or the NOAA) used in Switzerland. The use of different types of techniques can lead to significant errors. Could you justify with at least one example, the similitude of the results of both techniques?
- According to figure 3 and other comments in the discussion and the conclusions, how optimistic are you in the improvement of the nowcasting using this technique? And which could be the ways for improving the ML technique (e.g. other data sources, other thresholds, the ML itself) in the future?

Minor comments

- Figure 3: If the methodology (section 3) considers that Maximum reflectivity should exceed 37 dBZ, I cannot understand the Figure, because it shows that most of the time MaxZ do not reach this threshold and even in one case it never exceeds this value.
- In the same way that the previous point: which is the reason of selecting a so low reflectivity threshold (37 dBZ), considering that severe thunderstorms present values much higher than this threshold. Could you explain the motivation of your choose?
- Figure 4: I assume that as higher is the value of y-axis, worst is the performance. But, how do you really quantify the quality of the performance? E.g. POD values close to 0 (1) are very bad (good) skill values, or the opposite, FAR close to 1 (0) indicates bad (good) performance.
- Paragraph of L290: how good do you assume is your performance in operational terms with an increase of the MAE of 1.2 dB. Can you explain it?
- The occurrence of hail is poorly dependent of the occurrence of 45 dBZ, because of different reasons: values are concentrated at low levels, or the freezing height is much higher than the EchoTop45. In my opinion, choosing VIL parameter gives a better correlation (also poor, but less in any case) with hail occurrence. I would like that you provide a clarification of your selection
- Paragraph of L320: The increase of the influence of the NWP in time over the forecast is a well-known fact. Could you be more concise in the weight of this data source in your results? How do you explain the "valleys" in the relationship with radar data? (Fig. 6)

---

## Author Response (AR1)

Dear Editor and Reviewers,

Thank you for your comments on our manuscript "Nowcasting thunderstorm hazards using machine learning: the impact of data sources on performance". Attached, we submit the version that has been revised following the reviewer comments.

We have made several changes to the original manuscript in response to the reviewer comments. For our response to each comment of the reviewers, and the details of the changes made to the manuscript in response to each comment, please refer to the responses posted to each comment in the interactive discussion. You can also find the details of the changes made in the accompanying track-changes file.

In addition to changes made in response to the reviewer comments, we updated the revised manuscript with a few minor grammatical and typographical corrections that we found appropriate during the revision process. We also made the machine-learning dataset and code used in the paper publicly available at this stage, and added the information pointing to it in the "Code and data availability" section at the end of the article.

Best Regards,

Jussi Leinonen (on behalf of all the authors)